# Current Roles of PET/CT in Thymic Epithelial Tumours: Which Evidences and Which Prospects? A Pictorial Review

**DOI:** 10.3390/cancers13236091

**Published:** 2021-12-03

**Authors:** Filippo Lococo, Marco Chiappetta, Elizabeth Katherine Anna Triumbari, Jessica Evangelista, Maria Teresa Congedo, Daniele Antonio Pizzuto, Debora Brascia, Giuseppe Marulli, Salvatore Annunziata, Stefano Margaritora

**Affiliations:** 1Università Cattolica del Sacro Cuore, 00168 Rome, Italy; marco.chiappetta@policlinicogemelli.it (M.C.); stefano.margaritora@policlinicogemelli.it (S.M.); 2Thoracic Surgery, Fondazione Policlinico Universitario A. Gemelli IRCCS, Università Cattolica del Sacro Cuore, Largo F.Vito 1, 00168 Rome, Italy; Jessica.evangelista@guest.policlinicogemelli.it (J.E.); mariateresa.congedo@policlinicogemelli.it (M.T.C.); 3Section of Nuclear Medicine, University Department of Radiological Sciences and Hematology, Università Cattolica del Sacro Cuore, 00168 Rome, Italy; elizakat@virgilio.it; 4Department of Nuclear Medicine, University Hospital Zurich, 8091 Zurich, Switzerland; danieleantonio.pizzuto@guest.policlinicogemelli.it; 5Unità di Medicina Nucleare, TracerGLab, Dipartimento Diagnostica per Immagini, Radioterapia Oncologica ed Ematologia, Fondazione Policlinico A. Gemelli IRCCS, 00168 Roma, Italy; salvatore.annunziata@policlinicogemelli.it; 6Unit of Thoracic Surgery, University of Bari, 70126 Bari, Italy; debora.brascia@uniba.it (D.B.); beppemarulli@libero.it (G.M.)

**Keywords:** ^18^F FDG PET/CT, thymoma, thymic epithelial tumour, radiometabolic assessment, WHO, histology

## Abstract

**Simple Summary:**

Thymic epithelial tumours are uncommon malignancies. Histologically, they may be distinguished in different subtypes and different relapse risk classes. Surgery, sometimes after induction therapy, stays the best treatment option, and long-term results depend on the disease stage and completeness of resection. In this context, ^18^F FDG PET CT scan has been reported to play different roles in the care strategy of thymic epithelial tumours. In the present review, we analyse current evidences, the use of this imaging tool and future application prospects.

**Abstract:**

Background: The use of ^18^F FDG PET/CT scan in thymic epithelial tumours (TET) has been reported in the last two decades, but its application in different clinical settings has not been clearly defined. Methods: We performed a pictorial review of pertinent literature to describe different roles and applications of this imaging tool to manage TET patients. Finally, we summarized future prospects and potential innovative applications of PET in these neoplasms. Results: ^18^FFDG PET/CT scan may be of help to distinguish thymic hyperplasia from thymic epithelial tumours but evidences are almost weak. On the contrary, this imaging tool seems to be very performant to predict the grade of malignancy, to a lesser extent pathological response after induction therapy, Masaoka Koga stage of disease and long-term prognosis. Several other radiotracers have some application in TETs but results are limited and almost controversial. Finally, the future of PET/CT and theranostics in TETs is still to be defined but more detailed analysis of metabolic data (such as texture analysis applied on thymic neoplasms), along with promising preclinical and clinical results from new “stromal PET tracers”, leave us an increasingly optimistic outlook. Conclusions: PET plays different roles in the management of thymic epithelial tumours, and its applications may be of help for physicians in different clinical settings.

## 1. Introduction

Thymic epithelial tumours (TETs) arise from the thymus and represent approximately 0.2–1.5% of all malignancies [1,2,3]. The World Health Organization’s histological classification is based on morphology, atypia grade and lymphocytic presence [4,5], and represents an independent prognostic factor for survival in these patients [6,7]. Considering histological characteristics and survival outcome, it is possible to categorize TETs in different risk classes, with a low risk class including types A, AB, and B1, and a high risk class including B2, B3, and C thymic neoplasms [8], even if for type C thymomas (thymic carcinomas) a clearly different survival outcome is present compared to thymomas [9].

Surgical treatment is the primary indication in presence of TETs, and it is indicated considering the radiological characteristics only, without need of pre-operative biopsy. On the other hand, information about histology may be useful to indicate integrated treatments such as neoadjuvant therapy in case of advanced thymomas/thymic carcinomas, or to suggest an appropriate surgical approach in case histology indicates a high risk of neighbouring structure infiltration [10,11,12].

TETs are usually studied using computed tomography (CT) and magnetic resonance imaging (MRI) to categorize mediastinal lesions [13,14], but these two instruments present limitations to distinguish between the histological subtypes of TETs [15,16].

Regarding this point, the use of fluorine-18-fluorodeoxyglucose (^18^F-FDG) positron emission tomography (PET) and PET/CT rapidly increased for the evaluation of TETs. [17,18] Indeed, PET image analysis may give peculiar information using not only a qualitative (visual) method, but also a semi-quantitative method, such as the calculation of the maximum standardized uptake value (SUVmax) [19]. Furthermore, the use of PET/CT scanners permits to combine metabolic information from PET with morphological data from CT [19].

However, two different main questions emerge regarding the clinical value of the ^18^F -DG PET in TET:Which role has PET for diagnosis, staging, response prediction to treatment and long-term prognosis today?Which are the actual criticisms and future perspectives in this field?

We performed a pictorial review of pertinent literature to offer a “panoramic view” on these topics.

## 2. The Role of ^18^F-FDG PET/CT in Thymic Epithelial Tumours

### 2.1. PET/CT to Distinguish Thymic Hiperplasia from Thymic Epithelial Tumours

#### ^18^F-FDG PET May Be Useful to Distinguish Thymic Hiperplasia from Thymic Epithelial Tumours Is Integrated with Anatomical Consideration and Spatial ^18^F-FDG Uptake Distribution

The role of ^18^F-FDG PET to distinguish benign from malignant thymic lesions, considering only the standardized uptake values (SUV), could be limited. In this regard, false positive findings may occur despite that the negative predictive value of PET/CT in evaluating mediastinal masses was high.

Indeed, patients with thymoma usually present selected and limited areas of increased FDG uptake (see a descriptive example in Figure 1), while in thymic lymphatic hyperplasia (TLH) the uptake distribution pattern may be more diffuse [20,21].

However, focal areas of FDG uptake may be found in some cases of TLH [20]. On the other hand, it is important to note that FDG uptake occurs physiologically in the thymus and may be a common finding in children and adults <40 years old [20,22]. The gland’s morphology may also be useful to distinguish a normal thymus from TLH. A V-shape morphology or a triangular anterior pre-vascular mass is the classical presentation of a normal thymus [20]. The SUVmax evaluation results controversial, even if El Bawad et al. [21] reported a significant difference comparing this parameter in TLH and thymomas. In other experiences, TLH and thymomas were reported with overlapping SUVmax results, especially in Masaoka Koga stages I-II or A-AB types [21,23]. However, in TLH, the SUVmax rarely was >3, which may be considered as a threshold for TLH, as shown by Watanabe et al. [23] and El Bawad et al. [21], who reported an upper limit of the SUVmax value detected in TLH of 3 and 3.2, respectively.

### 2.2. PET/CT to Stage Thymic Epithelial Tumours

#### PET/CT Permits a Discrete Stage Prediction in TETs

Correct identification of the clinical stage remains a fundamental part and milestone for treatment planning in TETs management [20]. Indeed, the characteristics of the tumour led physicians to plan integrated treatments such as induction chemo-/radiotherapy or the addition of hyperthermic intrathoracic chemotherapy (HITHOC) during surgery [24,25,26,27]. Local tumour characteristics are usually investigated using contrast CT or magnetic resonance, while ^18^F-FDG PET/CT is especially useful to detect metastases or to assess lymph node involvement [24,25]. Several studies considered the potential role of ^18^F-FDG PET/CT in TET staging, reporting interesting results (see Table 1).

The first studies began in the 1990s [28,29], suggesting a possible role of ^18^F-FDG PET (/CT) to discriminate non-invasive vs. invasive TET, but more recent studies deeply analysed this possibility by investigating metabolic parameters other than SUV. In detail, Ito and colleagues [30] analysed the association between FDG uptake, Masaoka Koga [31] and TNM stages [32]; they reported a significant difference in SUVmax comparing Masaoka Koga stages III-IV vs. I-II and identified an optimal cut-off of 5.4. Similarly, the authors reported a significant difference comparing TNM stages III-IV vs. I-II and identified an optimal cut-off of 5.6. Comparable results were described by Matsumoto et al. [33], who reported a significantly higher SUVmax for Masaoka Koga stage IV compared to stages I-II and for large tumours (>60 mm) compared with small tumours (<60 mm). Conversely in the study by Fukumoto et al. [34], the SUV difference comparing Masaoka Koga stage III-IV vs. I-II raised statistical significance (*p* = 0.06). Similarly, Lee et al. [29] investigated the role of SUVmax, average SUV, metabolic tumour volume (MTV) and total glycolytic volume (TGV), pointing out a significant difference comparing these parameters in the different Masaoka Koga stages. However, other studies did not report significant associations between metabolic parameters and TET staging, or reported important limitations. Bertolaccini et al. [35] investigated the potential role of MTV, TGV and SUV tumour/mediastinum (SUV T/M) ratio, and assessed only a weak statistically significant association between SUV T/M and Masaoka Koga stages. These results are in line with other studies that did not find stage prediction considering metabolic PET/CT parameters [36,37].

Watanabe et al. reported a statistically significant SUVmax difference considering Masaoka Koga stage III vs. I and TNM stage T3 vs. T1a, but the author also described the presence of extensive SUVmax distribution overlapping in WHO histological thymoma types and Masaoka Koga stage thymomas [23], as also outlined by Matsumoto et al. [33].

However, it is also important to note that other studies highlighted the better performance of ^18^F-FDG PET/CT to detect disease recurrence and distant metastases than CT [37,38].

Finally, there are no clear data on the association between PET/CT findings and the completeness of resection. As reported above, tumour-size seems to be associated with PET/CT findings. This probably could have an impact on the rate of R+-resection. Further analysis should be performed on this specific topic. In summary, the role of ^18^F-FDG PET for staging purposes remains controversial, mainly owing to the lack to define SUV cut-offs for different stages despite that the association between high SUV values and invasive TETs is quite clear. In our opinion, ^18^F-FDG PET/CT may be used for two main staging goals:for distant localizations or lymph node involvement assessment; andfor invasive thymoma identification.

Distant metastases from TETs mostly involved pleural cavity, diaphragm, lung and more rarely extra-thoracic areas, such as intra-ocular region, spleen, liver, pancreas, spinal cord and bone [38]. As far as we know, no cohort study is available about this topic owing to the extremely rare occurrence of extrathoracic metastatic spreading.

Regarding point 2, patients presenting a high SUV value may be examined for contiguous structure infiltration, which require pre-operative supplemental imaging (magnetic resonance), an open surgical instead of a minimally invasive approach or intraoperative preparation for extensive resections (pericardium, great vessels or lung).

### 2.3. PET/CT to Predict the Grade of Malignancy of Thymic Epithelial Tumours

#### PET/TC Predicts Ina Such Reliable Manner the Grade of Malignancy in TETs, Especially Considering a Low Grade vs. High Grade Thymoma or Thymic Carcinoma vs. Thymoma

The information about histology may be essential to plan the appropriate treatment programme in TETs. Therefore, different studies analysed and confirmed the correlation between ^18^F DG PET/CT and the clinical behaviour of TETs [17,39,40]. Several studies investigated in a specific manner the ability of ^18^F-FDG PET or PET/CT to predict the WHO classification [35,36,37,38,39,40,41] (see also Figure 2).

Considering the rarity of the disease, no definitive conclusions can be drawn regarding the ^18^F DG PET/CT and histology prediction. However, a meta-analysis performed in 2014 [42] showed interesting results reducing bias among 11 different studies [43]. In detail, the meta-analysis reported a pooled weighted mean difference (PWMD) of SUVmax between high-risk and low-risk thymomas of 1.2 (95%CI: 0.4–2.0), a PWMD of SUVmax between thymic carcinomas and low-risk thymomas of 4.8 (95%CI: 3.4–6.1) and a PWMD of SUVmax between thymic carcinomas and high-risk thymomas of 3.5 (95%CI: 2.7–4.3). Data of this study support that ^18^F-FDG PET/CT has a role to predict the grade of malignancy in TETs, showing clear differences among different histology.

Recently, other authors [44,45,46] reported similar studies supporting the correlation between metabolic parameters and histology in TETs. Zaho et al. observed a significant association between SUVmax, SUVmax/tumour size and histological WHO classification of TETs in a monocentric cohort of 81 patients [44]. Very similar results were reported a few years ago by Tomita et al. [45] suggesting that the ratio SUVmax/T was more useful than SUVmax alone to differentiate low-risk from high-risk thymomas.

Interestingly, other authors [47] observed that a delayed scanning (dual-time-point ^18^F-FDG PET/CT) could improve diagnostic power for high-risk TETs. Finally, when focusing the analysis on volume-dependent radiometabolic parameters, Park et al. [46] found that MTV and TLG showed no correlation with the WHO classification. Additionally, they also confirmed a significant relationship between SUVmax and WHO classification.

### 2.4. PET/CT to Predict Pathological Response after Induction Therapy

#### The Knowledge of Pathologic Response Is Still Limited, but PET/CT Showed Interesting and Promising Results in This Field

Few studies analysed the prognostic value of ^18^F-FDG PET/CT for post-induction treatment assessment, and the same principles of treatment response evaluation in inoperable advanced TET. In particular, RECIST and PRECIST criteria were adopted for pre and post treatment imaging evaluation [48], and it seems clear that a decrease of ^18^F-FDG PET/CT metabolic parameter should be associated with survival outcome especially considering progression free survival with improved outcome reported for responders compared with non-responders [48,49].

In detail, a decrease of almost 25–30% in SUVmax comparing pre- and post-therapy ^18^F-FDG PET/CT may be considered the cut-off for therapy response, even if Moon et al. [48] considered additionally other metabolic parameters stating that percent decrease of MTV and TLG resulted significant predictors of progression-free survival. The authors also individuated a decrease of 88.0% in MTV and 92.0% in TLG as the optimal cut-off value for disease progression.

Encouraging results of PET/CT’s prognostic role after induction therapy have been published, even if related to studies with a small number of patients. It is important to define pathological parameter of response, currently considered as follows: Ef0, no necrosis of tumour cells; Ef1, some necrosis of tumour cells with more than one-third of viable tumour cells; Ef2, less than one third of viable tumour cells; Ef3, no viable tumour cells viable (complete pathological response).

In a study on 14 patients, Fukumoto et al. reported no significant differences considering the variation in SUVmax and SUVindex in Ef0-Ef1 (no response) patients, while interesting findings were present in five of the six Ef2 patients [50]. In this subgroup of patients, SUVmax and SUVindex decreased in a range of 55–87% and 57–86% respectively. Moreover, RECIST criteria reported a stable disease in 3 of them. The only patient that did not present consistent SUV variation presented a very low pre-induction SUVmax of 2.9 (thymic carcinoma) [50]. The authors also analysed specificity and the sensibility of CT and ^18^F-FDG PET/CT for good responder individuation. Using CT, the area under the curve (AUC) was 0.667, setting the cut-off point at −27%, sensitivity and specificity were 88.3 and 62.5%, respectively. Using ^18^F-FDG PET with a 55% SUVmax rate reduction cut-off, the AUC was 0.896 with sensitivity and specificity of 88.3 and 100%, respectively.

Korst et al. described, in a cohort of 21 patients who underwent induction therapy, a radiologic tumour shrinkage according with the RECIST criteria in only 50% of the cohort, suggesting that only CT variations may be not enough for post-treatment evaluation [51]. However, five patients presented SUVmax normalization, and in the entire cohort, the median magnitude of SUVmax decrease resulted 44.5%. Five patients presented Ef2 response with a SUVmax decrease ranging between 39% and 100% (except for one type A thymoma patient who presented a SUVmax increase with 10% of viable tumour cells) suggesting the effectiveness of the ^18^F-FDG PET/CT for tumour response assessment after induction therapy.

Similarly, Matsumoto et al. noted ^18^F-FDG PET/CT effectiveness in therapeutic efficacy outcome in 5 patients with increased pathologic response associated with larger decreases in SUVmax [33].

In conclusion, pre-and post-induction therapy ^18^F-FDG PET/CT parameters seem to be correlated with pathological response, even if further studies with larger patient cohorts are required to confirm these preliminary results.

### 2.5. PET/CT to Predict Prognosis

#### PET/CT Results Seem to Correlate with Prognosis Prediction in TET after Surgery and after Radio-Chemio Therapy, Even If Further Studies Are Needed to Validate These Data

While Masaoka staging seems to be almost adequate to predict prognosis [11,24], this staging system is substantially based on surgical findings. For this reason, physicians need to explore pre-operative imaging tools to predict prognosis in thymic epithelial tumours. PET/CT findings have been partly studied in patients with thymic epithelial tumours, and most studies investigated the association between ^18^F-FDG PET/CT parameters and WHO classification or Masaoka stage [29,52,53,54]. Conversely, no results were reported regarding the prognostic value of ^18^F-FDG PET/CT considering the short follow-up.

In 2014, Seki et al. assessed the correlation between ^18^F-FDG PET/CT findings and long-term results in 37 patients surgically treated for TETs. In detail, a significant shorter recurrence-free survival was found in the group of TET-patients with SUVmax > 4.27) compared with those with SUVmax ≤ 4.27 (*p* = 0.0009) [55].

Another potential role of ^18^F-FDG PET/CT regards the possibility to analyse treatment response also considering the Response Evaluation Criteria in Solid Tumors (RECIST). Thomas et al. analysed the metabolic response in a cohort of unresectable Masaoka Koga stage III or IV TET patients who underwent non-surgical treatment, and found a close correlation between early metabolic response and subsequent best response according to RECIST criteria (*p* < 0.0001–0.0003). The author reported a sensitivity and specificity in best response prediction of 95% and 100%, respectively. Regarding survival correlation, patients with a metabolic response had significantly longer progression-free survival (PFS) and a trend towards longer overall survival than non-responders: PFS median 11.5 vs. 4.6 months (*p* = 0.044); OS median 31.8 vs. 18.4 months (*p* = 0.14) [49].

Moreover, Lee et al. [29] investigated the role of different metabolic parameters on disease free survival or PFS in TETs reporting that the average standardized uptake values resulted an independent prognostic factor. In this retrospective analysis of 83 TET patients, the volumetric PET/CT parameters were assessed according to a threshold of SUVmax 2.5. Among other factors (Masaoka Koga stage, histologic types and treatment modality), the SUVmax, average SUV (SUVavg), MTV and TLG were significant prognostic factors on univariate analysis, while multivariate analysis confirmed as independent prognostic factors the SUVavg (*p* < 0.001) and Masaoka Koga stage (*p* = 0.001).

The usefulness of PET/CT for detection of local recurrence in TETs patients after thymectomy is largely unexplored. A single retrospective analysis about the comparison of the accuracy of ^18^F-FDG PET/CT and CT scan alone in 37 patients suspected for recurrence after surgery was published. Briefly, the authors reported a PET/CT sensitivity and specificity of 82% and 95%, respectively, vs. 71% sensitivity and 85% specificity reached with CT scan alone. Particularly, a PET/CT sensitivity of 100% was reached in patients who presented post-surgical recurrence in the anterior mediastinum. The authors hypothesized that patients could benefit from hybrid imaging approach for detection and localization of post-surgical recurrence [37].

## 3. Future Perspectives

### 3.1. Other PET/CT Radiopharmaceuticals in Thymic Epithelial Tumours

With the exception of ^18^F-FDG, no further positron-emitter radiotracer has been introduced in the standard workup of patients with thymic neoplasms. However, some cases in literature described incidental thymic findings during non-^18^F-FDG PET/CTs.

A few case reports described incidental findings of an abnormal mediastinal ^18^F- [56], 11C- [57] or ^18^F -Fluoroethil- [58] Choline uptake in PET/CT scans of prostate cancer patients with biochemical recurrence. The following CT-guided biopsy and histopathological examination revealed a diagnosis of epithelial thymoma [56,57] or thymic carcinoma with pleural metastases [58]. Of note, the patient who underwent ^18^F -FECH-PET/CT was also subjected to ^18^F-FDG PET/CT, and, interestingly, mediastinal blood pool activity in the latter exam negatively affected tumour demarcation, while ^18^F -FECH distribution resulted in a high target to background signal ratio and better tumour delineation. Therefore, the authors raised the question about the possibility to use radiolabelled Choline PET/CT in standard diagnostic strategies for thymus cancer patients.

Another study reported the case of a patient with severe familial hyperparathyroidism who underwent an ^18^F -Fluorocholine PET/CT and was referred for surgery because of a mediastinal lesion suspect for being an ectopic parathyroid; instead, the lesion was pathologically classified as thymoma [59].

In a multicentre study by Shibata et al. [60], 40 patients were studied with ^18^F-FDG and 11C-labeled acetate (AC) PET/CT with the aim to predict histologic types and tumour invasiveness of thymoma. Results reported that neither the FDG-SUV nor the AC-SUV failed to predict the invasiveness of thymomas assessed by tumour stage. Conversely, these semi-quantitative parameters could be able to predict at least some histologic types: FDG-SUV < 6.3 and AC-SUV ≥ 5.7 almost certainly indicated types A/AB thymoma. This was considered to have substantial prognostic and management significance. Other authors came to the same conclusions after studying three patients with thymoma through 11C-AC PET/CT finding; one patient resulted false negative in an ^18^F-FDG PET/CT scan [61]. Sakurai et al. [62] described a similar case of strong 11C-AC uptake and blunt ^18^F-FDG uptake, which helped to diagnose ectopic -and tumorous- thymic tissue.

Another PET/CT tracer, is 68Ga-PSMA, which has been associated with thymic neoplasms and is usually adopted in prostate-cancer-patients. 68Ga-PSMA has been suggested for the assessment of other slow-growing tumours such as thymomas [63] or even more aggressive thymic carcinomas [64].

From the perspective of a potential therapeutic approach to thymomas, nuclear medicine has implemented radiopharmaceuticals, and, in the specific case of thymic lesions, only one experience on three patients with primary neuroendocrine tumour of the thymus has been described in literature [65]. In this study, nine metastatic or advanced and inoperable primary neuroendocrine tumours of rare sites were treated, eight out of nine with at least two cycles of 177Lu-DOTATATE. Among the three patients of interest for this review, two patients (one with thymic carcinoid and the other with an atypical carcinoid of the thymus) had a biochemical and clinically stable disease after treatment of a partial biochemical response and clinically stable disease in the second, respectively. The third patient, diagnosed with a typical carcinoid of the thymus with orbital and mediastinal metastases, had a partial clinical and biochemical benefit after the first two cycles of therapy, but progressed thereafter.

### 3.2. Future Perspectives

The future of PET/CT and theranostics in TETs is still to be defined, but some considerations can be made. As hybrid imaging could potentially play a key role for better staging and treatment response assessment in thymic neoplasms [66], further meta-analyses collecting available data of ^18^F-FDG PET/CT accuracy and more data about other hybrid imaging tools such as PET/MR could improve the management of patients with TETs, both to stage and to assess therapy response [66].

Furthermore, a more detailed analysis of metabolic data, such as texture analysis applied on thymic neoplasms, may ameliorate diagnostic performance and provide a more accurate prognostic stratification to ensure patient tailored treatments. [67].

New tracers may also be introduced in the management of TETs to target their stromal component using quinoline-based PET tracers that act as fibroblast activation protein inhibitors to individuate areas of overexpressed cancer-associated fibroblasts. The recent development of these tracers demonstrated promising preclinical and clinical results [68].

## 4. Conclusions

^18^F-FDG PET/CT scan may play different roles in the management of thymic epithelial tumours, and its applications may be of help for physicians in different clinical settings, to predict the grade of malignancy as well as to identify the risk class for relapse. The future of PET/CT and theranostics in TETs is still to be defined.

## Figures and Tables

**Figure 1 cancers-13-06091-f001:**
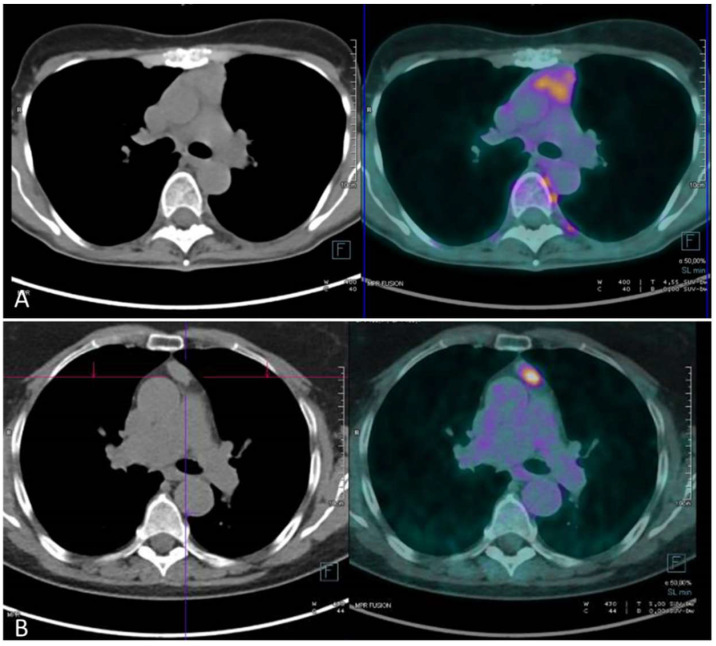
PET/CT findings in a patient with thymic hyperplasia (**A**) and a suspected thymoma (**B**). (**A**) a diffuse weak uptake is present in the anterior mediastinum. (**B**) an intense uptake (SUVmax: 5.4) was detected at the level of a mediastinal lesion; a B2-type thymoma was revealed after surgical resection.

**Figure 2 cancers-13-06091-f002:**
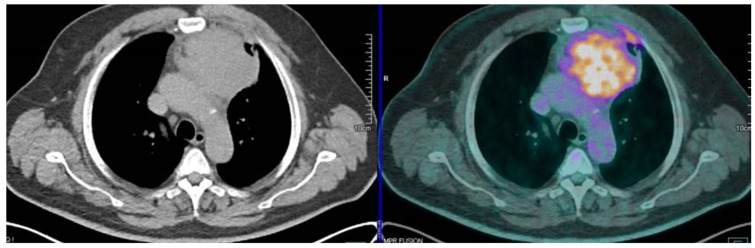
PET/CT findings in a patient with thymic carcinoma, presenting an elevate SUVmax of 12.8.

**Table 1 cancers-13-06091-t001:** Association between SUV max and Masaoka-Koga, TNM and histology in thymic epithelial tumors.

Study	Number of Patients	Masaoka Stage and ^18^F FDG SUV Max	Tnm Stage and ^18^F FDG SUV Max	Histology and ^18^F FDG SUV Max
**Sasaki (1999)**	29			Thymica carcinoma: 7.2 ± 2.9 Invasive thymoma: 3.8 ± 1.3 non-invasive thymoma: 3.0 ± 1.0 thymic cysts: 0.9 (*p* < 0.01)
**Matsumoto (2013)**	39	Masaoka stage I: 3.0 ± 1.1 Masaoka stage II: 3.6 ± 1.4 Masaoka stage III: 5.3 ± 3.4 Masaoka stage IV: 8.3 ± 4.0		A: 2.3 AB: 2.3 ± 0.8 B1: 3.7 ± 1.5 B2: 4.1 ± 1.4 B3: 4.9 ± 1.5 C: 12.1 ± 3.2 C vs. other histology *p* < 0.0001
**Bertolaccini (2014)**	23	Significant SUV T/M ratio Masaoka stage I: 2.31 ± 1.06 Masaoka stage II: 2.15 ± 1.27 Masaoka stage III: 2.69 ± 0.62 Masaoka stage IV: 2.53 (*ρ* = 0.555); no statistical correlation with MTV Masaoka stage I: 6.50 ± 3.69 Masaoka stage II: 5.40 ± 2.71 Masaoka stage III: 9.22 ± 1.86 Masaoka stage IV: 4.2 (*ρ* = 0.185), no statistical correlation with TGV Masaoka stage I: 233.7 ± 234.61 Masaoka stage II: 144.14 ± 276.18 Masaoka stage III: 431.80 ± 343.31 Masaoka stage IV: 55 (*ρ* = 0.199)		TGV and WHO classification: Low risk thymoma 99.12 ± 125.98 High risk thymoma 645.83 ± 159.87 (*q* = 0.897) SUV T/M ratio and WHO classification: Low risk thymoma 1.91 ± 0.45 High risk thymoma 3.73 ± 0.95 (*q* = 0.873).
**Watanabe (2018)**	63	Masaoka stage I: 3.0 ± 1.2 Masaoka stage II: 3.8 ± 1.6 Masaoka stage III: 4.4 ± 1.2 Masaoka stage IV: 4.8 ± 1.0	T1a: 3.2 ± 1.3 T1b: 5.4 ± 2.0 T2: 5.1 ± 0.9 T3: 4.6 ± 1.1 T3 and T1b thymomas had significantly higher SUVmax than T1a	A: 3.1 ± 1.8 AB: 3.4 ± 1.5 B1: 3.8 ± 1.5 B2: 3.7 ± 1.6 B3: 4.3 ± 1.0 No significant difference
**ITO (2020)**	56	Masaoka stage I-II: 4.52 ± 2.66 Masaoka stage III-IV: 7.73 ± 2.83 (*p* < 0.01) Cut off: 5.6 (sensitivity 0.89, specificity 0.78)	T1a: 4.45 ± 2.06 T1b: 4.9 ± 2.0 T2: 7.12 ± 2.69 T3: 8.31 ± 2.57 T4: 9.79 ± 7.48 T3 vs. T1a (*p* < 0.01)	Thymic carcinoma vs. high-grade thymoma: 9.09 ± 3.34 vs. 6.01 ± 2.78 (*p* < 0.01), Cut-off 7.40 (sensitivity 0.72, specificity 0.79) high-grade thymoma vs. low-grade thymoma: 6.01 ± 2.78 vs. 4.06 ± 1.86 (*p* < 0.01) Cut-off 5.40 (sensitivity 0.61, specificity 0.85
**LEE (2021)**	83	Masaoka stage I: 6.6 ± 4.8 Masaoka stage II: 4.9 ± 1.8 Masaoka stage III: 8.9 ± 5.7 Masaoka stage IV: 10.5 ± 6.7 (*p* = 0.001)		Low risk thymomas 3.98 ± 2.59 high risk thymoma 5.63 ± 2.89 thymic carcinoma 10.42 ± 3.53 *p* < 0.001

T stage based on the TNM 8th edition. SUV T/M ratio: standardized uptake value of tumour/standardized uptake value of mediastinum, MTV: metabolic tumour volume, TGV: total glycolytic volume.

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
