# Peer review of "Current Roles of PET/CT in Thymic Epithelial Tumours: Which Evidences and Which Prospects? A Pictorial Review"

_cancers, 2021, doi:10.3390/cancers13236091_

Round 1
Reviewer 1 Report
The authors have largely addressed my remarks. I would propose adding more figures of case examples such as Figure 1 to make it more illustrative to the readership.
Author Response
COMMENT: The authors have largely addressed my remarks. I would propose adding more figures of case examples such as Figure 1 to make it more illustrative to the readership.
ANSWER: thank you for your suggestion. We added other figures hoping that may be useful as examples of PET outcome in different thymic pathology.
Reviewer 2 Report
In table 1, there are some typos. Furthermore, if the authors separate this table to tumor stage and malignancy (benign, thymoma, invasive, carcinoma), the readers will more easily understand the results. It is most important for us to predict the grade of malignancy of TET, thus more detailed results should be described in the table. Could the authors describe the simple summary in each section? I recommend revising the manuscript so that the clinical values and usefulness of PET/CT in treatment of TET appear more clearly.Author Response
QUESTION: In table 1, there are some typos. Furthermore, if the authors separate this table to tumor stage and malignancy (benign, thymoma, invasive, carcinoma), the readers will more easily understand the results. It is most important for us to predict the grade of malignancy of TET, thus more detailed results should be described in the table. Could the authors describe the simple summary in each section? I recommend revising the manuscript so that the clinical values and usefulness of PET/CT in treatment of TET appear more clearly.
REPLY: Thank you for your suggestion. We modified the table dividing the outcome based on Masaoka kga, TNM and histology. We also integrate the paragraphs with a short summary at the beginning. We hope that these integrations improved the manuscript
This manuscript is a resubmission of an earlier submission. The following is a list of the peer review reports and author responses from that submission.
Round 1
Reviewer 1 Report
The manuscript describes the value of FDG PET/CT in thymic epithelial tumors as systemic reviews. The topic is of clinical importance and contemporarily relevant since there is a paucity in reports on the link between thymoma and FDG uptake, which is limited to a few studies with low incidence. However, there are several questions regarding the design of this study.
- Introduction, Materials and Methods, and Results
1) The methodology is flawed by the unclear inclusion criteria and irrelevant methodologies. To investigate clinical performance of an imaging as meta-analysis, it is essential to establish an appropriate collected articles without bias. In this study, it is questionable whether the references were collected relevantly. Please show a flow chart showing the study selection process, Quality assessment, and Qualitative/quantitative synthesis.
2) It is difficult to figure out the main findings of the present study. Please categorize the benefits from FDG PET/CT and more details are needed how the clinical values were added. Table of the study/patients characteristics of the included studies would help readers have a better understanding.
2. 2.3, L16-173What is the definition of “robust evidence” for the cut-off of SUVmax? It does not appear to be informative number of 3.5 to compare benign lesions as forehead mentioned of 3.0 as benign. It is requested to describe in detail how the differential diagnosis was made using FDG PET CT findings.
3. 2.3 L182-184 The limitation of the WHO classification could not reflect on prognosis precisely. Masaoka stage has been introduced and used most widely to determine further treatment and to predict prognosis. It is recommended to add the relevant content between WHO classification, Masaoka stages, and FDG PET/CT.
4. It is difficult to figure out the difference of the ‘predict pathologic response’ and ‘predict response of treatment’ sections. Please categorize the unified contents from treatment response and prognosis.
5. Minor issues
1) Please correct superscripts of the FDG PET/CT across the paper.
2) How about description of the name of tracer at the ABSTRACT because the meaning of PET/CT is just filmmaking technology. It is better to insert tracer name at KEYWORDS in the same vein.
3) The abbreviation should be descripted of TLH.
4) It would be better to remove older references and replace with more contemporary references.
Author Response
Reviewer #1 Comments for the Author:
The manuscript describes the value of FDG PET/CT in thymic epithelial tumors as systemic reviews. The topic is of clinical importance and contemporarily relevant since there is a paucity in reports on the link between thymoma and FDG uptake, which is limited to a few studies with low incidence.
Reply: Thanks for such nice words. We will edit the manuscript in order to further improve the overall quality of the manuscript
However, there are several questions regarding the design of this study.
- Introduction, Materials and Methods, and Results
1) The methodology is flawed by the unclear inclusion criteria and irrelevant methodologies. To investigate clinical performance of an imaging as meta-analysis, it is essential to establish an appropriate collected articles without bias. In this study, it is questionable whether the references were collected relevantly. Please show a flow chart showing the study selection process, Quality assessment, and Qualitative/quantitative synthesis.
Reply: Thanks for such proper comment. Actually, there is a misunderstading concerning the real aim of the present study. Indeed, as correctly remarked by the Reviewer “..to investigate clinical performance of an imaging as meta-analysis, it is essential to establish an appropriate collected articles without bias”.
However, the present study stays as a kind of “overview” of the literature with the final aim of presenting a “Pictorial Review” of the role of PET/CT scan in thymic epithelial tumors.
Indeed, we have observed from previous studies that no strong evidences are available to perform a meta-analysis on the topics considered in the present study. Thus, we prefer to offer to the Readers a descriptive and panoramic essay. Consequently, we have not adopted the standard research methology (i.e. PRISMA) of a meta-analysis, but we selected those refernces considered as most representative (i.e. large cohort size) of the topic analysed.
To better clarified our intention, we have modified the title, the abstract and the introduction.
2) It is difficult to figure out the main findings of the present study. Please categorize the benefits from FDG PET/CT and more details are needed how the clinical values were added. Table of the study/patients characteristics of the included studies would help readers have a better understanding.
Reply: Thanks for your constructive comment. We completely agree with your suggestion and we have tried to simplify and summarized the results of the studies analysed by including in the manuscript a Table reporting studies on TETs staging and PET/CT findings. We hope these could be of help in manuscript reading and understanding.
- 2.3, L16-173What is the definition of “robust evidence” for the cut-off of SUVmax? It does not appear to be informative number of 3.5 to compare benign lesions as forehead mentioned of 3.0 as benign. It is requested to describe in detail how the differential diagnosis was made using FDG PET CT findings.
Reply: Thank you for your comment but there is a wrong interpretation of the manuscript. In the paragraph relative to the Treglia’s metanalysis it is not reported a SUVmax cut off, but the pooled weighted mean difference between low grade and high grade thymomas or thymic carcinomas. The paper clearly reported proportional suv max different among different histology.
- 2.3 L182-184 The limitation of the WHO classification could not reflect on prognosis precisely. Masaoka stage has been introduced and used most widely to determine further treatment and to predict prognosis. It is recommended to add the relevant content between WHO classification, Masaoka stages, and FDG PET/CT.
Reply: Thanks for such meaningful consideration. Actually, as You correctly stated WHO classification could not reflect on prognosis precisely while Masaoka seems to be more adequate in this setting, despite this staging is substantially based on surgical findings. For this reason, physicians need to explore pre-operative imaging tool to predict prognosis in thymic epithelial tumors. PET/CT findings have been partly studied in patients with thymic epithelial tumors. Although a previous study showed an association of volume-dependent 18F-FDG PET/CT parameters with proposed prognostic factors, including WHO classification and Masaoka stage [24], there were no results regarding the prognostic value of 18F-FDG PET/CT due to the short follow-up duration.
This content was included in the beginning of chapter 2.5 in te context of a re-editing of this part of the manuscript to follow your suggestion.
- It is difficult to figure out the difference of the ‘predict pathologic response’ and ‘predict response of treatment’ sections. Please categorize the unified contents from treatment response and prognosis.
Reply: Thanks again for your relevant comment. Basically, we wanted to draft two different paragraphs: the first one for assessing the pathological response after induction therapy, the second one to describe the prognostic value of PET after any kind of treatment (surgery, CT, etc.). We have better defined the title of this two chapters and re-edited them accordingly.
- Minor issues
1) Please correct superscripts of the FDG PET/CT across the paper.
Reply: Done
2) How about description of the name of tracer at the ABSTRACT because the meaning of PET/CT is just filmmaking technology. It is better to insert tracer name at KEYWORDS in the same vein.
Reply: Done
3) The abbreviation should be descripted of TLH.
Reply: Done
4) It would be better to remove older references and replace with more contemporary references.
Reply: Some refernces are not old but historical.
For example
- Rosai J, Sobin LH. Histological typing of tumors of thymus. International histological classification of tumors, 2nd ed., New York: Springer; 1999.
This is the pubblication of the historical classification and deserves a mention even if reported in 1999.
An other example
- Kondo K, Yoshizawa K, Tsuyuguchi M, Kimura S, Sumitomo M, Morita J, Miyoshi T, Sakiyama S, Mukai K, Monden Y. WHO histologic classification is a prognostic indicator in thymoma. Ann Thorac Surg. 2004;77:1183-8
This is an impressive historical case series that firstly reported a large cohort of TETs-patients. Despite from 2004, the other more recent series had not the same “impact” on the audience of thoracic surgeons.
Thus, we have change some old references (7,41) before 2000, but we prefer to not change the other old references for the above-mentioned reasons.
Reviewer 2 Report
In their manuscript the authors present a narrative of the relevant review with the objective of clarifying the different roles and applications of this imaging tool to manage TETs patients, as well as to future prospects and potential innovative applications of PET in these neoplasms.
- This is a somewhat interesting review which however brings little new information to the literature.
- Moreover the manuscript is rather poor in tables and figures, which should definately be present given this is a review focusing on imaging studies.
- The information and data given might be accurate and up-to-date yet are not clearly presented to the readership.
- More data could be added on any eventual RCT running or registries? in the future perspectives section.
Author Response
Reviewer #2 Comments for the Author:
In their manuscript the authors present a narrative of the relevant review with the objective of clarifying the different roles and applications of this imaging tool to manage TETs patients, as well as to future prospects and potential innovative applications of PET in these neoplasms.
This is a somewhat interesting review which however brings little new information to the literature.
Reply: Thanks for such nice words. We will edit the manuscript in order to further improve the overall quality of the manuscript
Moreover the manuscript is rather poor in tables and figures, which should definately be present given this is a review focusing on imaging studies.
The information and data given might be accurate and up-to-date yet are not clearly presented to the readership.
Reply: Thanks for your constructive comment. We completely agree with your suggestion and we have tried to simplify and summarized the results of the studies analysed by including in the manuscript a Table reporting studies on TETs staging and PET/CT findings. We hope these could be of help in manuscript reading and understanding. Moreover a pictorial image of PET/CT in a thymoma has been added to the manuscript
More data could be added on any eventual RCT running or registries? in the future perspectives section.
Reply: Thanks for your suggestion. Actually, we are preparing an other manuscript exaclty on this topic and we prefer to not duplicate information. We hope you could understand this point.
Reviewer 3 Report
Congratulations on your important and interesting paper. I have a few suggestions which may help it to look even better.
- Add at the end of the Introduction a few key questions or goals of the paper you are trying to elucidate, eg: Is there any connection between the FDG uptake and R1 resection etc. Answering these questions at the end of the paper or saying that there are no exact answers will make the paper quite appealing in terms of the future prospects of research.
- I suggest adding a paragraph on how PET/CT may help in establishing recurrence after thymectomy if any of such information is available
- Discuss how predictable PET could be in achieving R0 results.
- To present data or at least comment on how effective PET in the detection of distant metastases in patients with thymic tumours.
Author Response
Reviewer #3 Comments for the Author:
Congratulations on your important and interesting paper. I have a few suggestions which may help it to look even better.
Reply: Thanks for such nice words. We will edit the manuscript in order to further improve the overall quality of the manuscript
Add at the end of the Introduction a few key questions or goals of the paper you are trying to elucidate, eg: Is there any connection between the FDG uptake and R1 resection etc. Answering these questions at the end of the paper or saying that there are no exact answers will make the paper quite appealing in terms of the future prospects of research.
Reply: Thanks for your suggestion. We have included a couple of topic (reported as questions) that the pectorial review wants to explore. We hope this version could be more appealing for the readers
I suggest adding a paragraph on how PET/CT may help in establishing recurrence after thymectomy if any of such information is available
Reply: Thanks for your comment. The usefulness of PET/CT for detection of local recurrence in TETs patients after thymectomy is largely unexplored. We have included a comment on this topic analysing the few data available in lietarture
“The usefulness of PET/CT for detection of local recurrence in TETs patients after thymectomy is largely unexplored. A single retrospective analysis about the comparison of the accuracy of 18F-FDG PET/CT and CT scan alone in 37 patients suspected for recurrence after surgery was published. Briefly, the authors reported a PET/CT sensitivity and specificity of 82% and 95% vs 71% and 85% reached with CT scan alone, respectively. Particularly, a PET/CT sensitivity of 100% was reached in patients who presented post-surgical recurrence in the anterior mediastinum. The authors hypothesized that patients could benefit from hybrid imaging approach for detection and localization of post-surgical recurrence [37].”
Discuss how predictable PET could be in achieving R0 results.
Reply: Thanks for your intruiging suggestion. We have extensively explored these data from literature but there are no paper focused on this topic. We have found only data on te correlation between tumor-size and PET/CT findings, this probably could have an impact on the rate of R+-resection. We have included this content in the manuscript
“Finally, there are no clear data on the association between PET/CT findings and the completeness of resection. As reported above, tumor-size seems to be associated with PET/CT findings, this probably could have an impact on the rate of R+-resection. Further analysis should be performed on this specific topic.”
To present data or at least comment on how effective PET in the detection of distant metastases in patients with thymic tumours.
Reply: Thanks for such comment. A brief comment has been included as required
“Distant metastases from TETs were reported to mostly involve pleural cavity, diaphragm, lung and more rarely extra-thoracic areas, such as intra-ocular region, spleen, liver, pan-creas, spinal cord, and bone [38]. As far as we know, not any cohort studies are available about this topic, owing to the extremely rare occurrence of metastatic spreading.”